# Genipin Attachment of Conjugated Gold Nanoparticles to a Decellularized Tissue Scaffold

**Mitch Bellrichard [1], Colten Snider [2], Cornelia Dittmar [2], John Brockman [3], Dave Grant [2] and Sheila A. Grant [2,\***

[1] Department of Veterinary Pathobiology, University of Missouri, Columbia, MO 65211, USA; bellrichardm@mizzou.edu

[2] Department of Biomedical, Biological, and Chemical Engineering, University of Missouri, Columbia, MO 65211, USA; clsytc@mail.missouri.edu (C.S.); cadyt7@mail.missouri.edu (C.D.); grantdav@missouri.edu (D.G.)

[3] Research Reactor University of Missouri, Columbia, MO 65211, USA; BrockmanJD@missouri.edu

\* Correspondence: GrantSA@missouri.edu

**Abstract:** Decellularized allograft tissue is used for a wide array of tissue injuries and repair with tenons and ligament repair being among the most common. However, despite their frequent use there is concern over the lengthy inflammatory period and slow healing associated with allografts. One promising solution has been the use of nanoparticles. There is currently no easy, fast method to achieve consistent conjugation of nanoparticles to tissue. The available conjugation methods can be time-consuming and/or may create numerous cytotoxic byproducts. Genipin, a naturally derived crosslinking agent isolated from the fruits of *Gardenia jasminoides* was investigated as a conjugation agent to achieve fast, consistent crosslinking without cytotoxic byproducts. The rational of utilizing genipin is that is reacts spontaneously with amino-group-containing compounds such as proteins, collagens, and gelatins, and does not require extensive washing after conjugation. Porcine diaphragm tendons were decellularized and then immersed in cysteamine functionalized gold nanoparticles and genipin for various time points. Tissue scaffolds were tested for the degree of crosslinking, gold nanoparticle concentrations, and fibroblast attachment and biocompatibility. Results demonstrated that genipin can successfully and reproducibly attach gold nanoparticles to tissue in as little as 15 min. The genipin had no cytotoxic effects and improved fibroblast attachment and proliferation. Genipin can be used to attach gold nanoparticles to tissue in a fast, cell safe manner.

**Keywords:** genipin; gold nanoparticles; cell adhesion; decellularized tissue; fibroblast

## 1. Introduction

Tissue engineering has advanced as a promising solution for the repair of damaged or diseased tissues with the goal of creating functional scaffolds that mimic native tissue and can be colonized by the host's cells. Decellularized tissue has shown promise in this regard, and there are numerous surgical scaffolds in clinic use today utilizing both allogenic and xenogeneic decellularized tissue including, urinary bladder, small intestine, dermis, mesothelium, heart valves, and pericardium [1]. In addition, researchers are working towards the use of three-dimensional scaffolds created through whole organ decellularization as a treatment for end-stage organ failure without the risk of chronic rejection and the morbidity associated with immunosuppression [2–4]. Biological tissue is better able to mimic the full complexity of the tissue architecture while also being a source of signaling molecules and growth factors creating a superb environment for cellular attachment and proliferation [5,6]. Although decellularized tissue has numerous promising characteristics, it is not without its drawbacks.

These concerns include decellularization weakening mechanical properties, inherent heterogeneity, high immunogenicity, rapid biodegradation, and slow integration [7–9]. This is specifically true with ligament grafts. Decellularized ligament and tendon use is limited due to a prolonged inflammatory period and delayed graft remodeling [10]. A potential solution to some of these concerns is the utilization of gold nanoparticles.

Gold nanoparticles (AuNPs), conjugated to decellularized tissue, can mitigate some of the concerns with decellularized tissue grafts. First, AuNPs have long been utilized for their anti-inflammatory properties, which is believed to be the result of free radical scavenging [11–13]. Secondly, gold nanoparticle attachment modify the surface structure and encourage cellular attachment and proliferation [14,15]. The increased surface energy of AuNPs may promote the attachment of proteins including those necessary for cellular attachment [16]. In addition, conjugation of the AuNPs to the tissue is believed to block collagenase attachment and thereby slow down scaffold degradation [15]. AuNPs can also be used to direct the differentiation of stem cells into specific cell types. Yi et al. demonstrated that AuNPs promote the osteogenic differentiation of mesenchymal stem cells by activating the p38 mitogen-activated protein kinase pathway, and Dong et al. demonstrated the addition of AuNPs promotes osteogenic differentiation of adipose-derived stem cells and results in significantly higher new bone formation in a rabbit model [17,18].

With all the promise of nanoparticles, their attachment to tissue remains challenging. The commonly used methods have cytotoxic byproducts and require multiple washes to remove them [19,20]. In addition, the current conjugation protocols only enable a rather rough estimate of the amount of attached gold as many of the agents only react for a limited period of time [21,22]. There is a need to find a biocompatible, stable, crosslinking agent to facilitate the attachment of nanoparticles. Genipin is a potential solution to this problem. Genipin is a chemical compound extracted from the fruits of *Gardenia jasminoides*. It is a natural crosslinking agent, and it spontaneously reacts with amino-group-containing compounds such as proteins, collagens, and gelatins to form mono- to tetramer crosslinks, and has an exceptionally low cytotoxicity [23,24]. Genipin also has the added benefit of acting as an anti-inflammatory agent and simultaneously reducing the immunogenicity of the scaffold [8,25,26].

In this study, it is hypothesized that genipin is a viable agent to conjugate gold nanoparticles to a decellularized tendon scaffold quickly and efficiently. A porcine diaphragm tendon was decellularized and exposed to solutions containing either genipin, 20 nm gold nanoparticles, or both for time points ranging from 15 min to 24 h. The thermal stability of the tissue was measured using differential scanning calorimetry to evaluate the degree of crosslinking and potential damage to the tissue. Neutron activation analysis (NAA) was used to measure the amount of gold attached to each sample. WST-1 and dsDNA assay PicoGreen was performed to measure viability of the fibroblast cells cultured on the scaffolds, and scanning electron microscopy was performed to visualize the fibroblast attachment.

## 2. Materials and Methods

### 2.1. Tissue Harvest and Decellularization

Porcine diaphragms were harvested immediately following euthanasia after a laboratory exercise at the University of Missouri. The central tendon portion of the diaphragm was dissected from the surrounding muscle. They were then decellularized according to a previously published protocol [22]. In brief, the tissues were immersed in a solution containing 1% (*v/v*) tri(n-butyl) phosphate (TnBP) (Sigma-Aldrich, St. Louis, MO, USA) in storage buffer solution and subjected to continuous agitation on an orbital shaker at ambient temperature for 24 h. The 1% TnBP solution was removed after 24 h and exchanged with fresh solution, and the tissues were subjected to continuous agitation for an additional 24 h. This treatment was followed by a 24 h rinse with double distilled water and another 24 h rinse with 70% (*v/v*) ethyl alcohol, both with continuous agitation at ambient temperature. This method of decellularization was previously studied in our lab and verified to effectively remove all cell nuclei

while leaving the structure and composition of the tissue intact [27]. 4.8 mm circular discs were cut from the decellularized diaphragm tendon and stored in 70% (*v/v*) ethanol at 4 °C.

### 2.2. Genipin, Gold Nanoparticles and the Crosslinking Procedure

Genipin (Sigma-Aldrich, St. Louis, MO, USA) crosslinking was conducted by immersing the decellularized tissue into 1 mL of solution containing 3 mM or 10 mM dissolved genipin. The genipin was dissolved using dimethyl sulfoxide and suspended in PBS. This was accompanied with 0.25 mL of 20 nm gold nanoparticles (Ted Pella, Redding, CA, USA) at a concentration of $7.0 \times 10^{11}$. The 20 nm AuNPs were utilized due to a previous study demonstrating the efficacy of nanoparticles in this size range in reducing inflammation [28]. Nanoparticles were functionalized with amine groups by the addition of 0.001 mg/mL 2-mercaptoethyamine (Cysteamine) (Sigma-Aldrich, St. Louis, MO, USA) to the nanoparticles. The scaffolds were crosslinked for 15 min, 1 h, 4 h or 24 h and then were quickly rinsed with PBS. The AuNP tissue control samples were created using the same methodology with the exception that the genipin solution was replaced with PBS.

Tissue scaffolds were sterilized following crosslinking by immersion in 90% ethanol for 24 h at 225 rpm. This was followed by three washes in sterilized phosphate buffered saline.

### 2.3. Experimental Groups

1. Untreated: porcine diaphragm tendon that underwent decellularization protocol.
2. 15 min, 1 h, 4 h, 8 h, 24 h Au Gen: decellularized tissue crosslinked with 0.25 mL of functionalized 20 nm gold nanoparticles at the stock and 1 mL of genipin at 3 mM. Crosslinking time ranged from 15 min to 24 h.
3. 15 min, 1 h, 4 h, 8 h, 24 h Gen: decellularized tissue crosslinked with 0.25 mL of PBSand 1 mL of genipin at 3 mM. Crosslinking time ranged from 15 min to 24 h. This group was crosslinked with genipin but without the addition of AuNPs.
4. 15 min, 1 h, 4 h, 8 h, 24 h Au: decellularized tissue crosslinked with 0.25 mL of functionalized 20 nm gold nanoparticles at the stock and 1 mL of PBS. Crosslinking time ranged from 15 min to 24 h. This group was conjugated with nanoparticles but without the addition of genipin.
5. EDC/NHS crosslinked tissue: decellularized tissue that were crosslinked with the chemical crosslinkers 1-ethyl-3-[3-dimethylaminopropyl]carbodiimide (EDC) (Thermo Fisher Scientific, Waltham, MA, USA) and N-hydroxysuccinimide (NHS) (Thermo Fisher Scientific, Waltham, MA) according to a previously published protocol [26]. Briefly, 2mM EDC 2-(N-Morpholino)ethanesulfonic acid buffer and combined with 5mM NHS in dimethyl formamide. These are then added to a 50:50 (*v/v*) acetone:phosphate buffered saline mixture. The tissue is incubated in 0.25 mL of crosslinking solution for 15 min. The tissue is then incubated overnight and then rinsed for 48 h in PBS at 225 rpm. EDC/NHS is a zero-length crosslinker commonly utilized to crosslinked tissue. It was utilized as a control in the DSC studies.

### 2.4. Neutron Activation Analysis

NAA was utilized to quantify the gold levels in the tissue scaffolds. Following crosslinking, five samples of each treatment type (N = 5) were lyophilized, weighed, and packed into high density polyethylene NAA vials. At the University of Missouri Research Reactor, the samples were loaded into a rabbit system with Au comparator standards and irradiated for 120 s in a thermal neutron flux of $5.0 \times 10^{13}$ n/cm$^2$/s. The $^{197}$Au captures a neutron to produce the radio-isotope $^{198}$Au with a 2.7 d half-life. The samples were allowed to decay for 1–7 h and then counted for 10 min each using a high purity Ge detector controlled by Canberra Genie 2000 software. The detector dead-time was less than 5% for all samples.

### 2.5. Modulated Differential Scanning Calorimetry

Following crosslinking, five specimens (n = 5) from the 7 treatment types were rinsed in DI water and placed in an aluminum pan with a hermetic lid. The treatment types were EDC/NHS, untreated, 15 min crosslinking with 3 mM genipin, 4 h crosslinking with 3 mM genipin, 8 h crosslinking with 3 mM genipin, 24 h crosslinking with 3 mM genipin, and 24 h crosslinking with 10 mM genipin. The 10 mM treatment type was added to verify that increasing the genipin concentration would further increase the amount of crosslinking that occurred. The reference pan consisted of an aluminum pan containing 2 μL of double distilled water and sealed with a hermetic lid. Each specimen was then subjected to modulated differential scanning calorimetry using a Q2000 DSC (TA Instruments, New Castle, DE, USA) to raise the temperature from 5 °C to 120 °C at a rate of 5 °C per minute with a modulation of ±0.64 °C every 80 s. The mean denaturation temperature is reported.

### 2.6. Cell Culture

L-929 mouse fibroblast cells were obtained from ATCC (Manassas, VA). They were cultured in EMEM (ATCC, Manassas, VA, USA) supplemented with 10% (*v/v*) horse serum (Sigma-Aldrich, St. Louis, MO, USA) and 200 U mL$^{-1}$ penicillin–streptomycin (Sigma-Aldrich, St. Louis, MO, USA) solution in an incubator at 37 °C and 5% $CO_2$. 1 mL of $3 \times 10^4$ cell/mL cell solution was plated on each scaffold and allowed to grow for 1, 3, 7, or 10 days with fresh media being replaced every 48 days.

### 2.7. Cell Viability

Cell proliferation reagent WST-1 (Roche Diagnostics Corporation, Indianapolis, IN, USA) was used to evaluate the biocompatibility of the scaffolds. The WST-1 assay works via the use of tetrazolium salts. The salts are added to wells containing cells and the tissue discs. The tetrazolium salts are then cleaved to formazan by mitochondrial dehydrogenase activity. This correlates to the number of metabolically active cells. The resulting formazan is can be quantified using UV-vis absorbance measurements. A total of 5 scaffolds (N = 5) from ten treatment types (untreated, 15 min, 1 h, and 24 h with 3 mM genipin and gold nanoparticles or either of the components independently) were seeded with L929 mouse connective tissue fibroblasts and incubated for 1, 3, 7, and 10 days with half of the media in each well replaced every 48 h. WST-1 reagent was added to each well and the plates incubated at 37 °C for 4 h. After gentle mixing, 100 μL was removed from each well and absorbance readings were acquired using a Tecan Safire II plate reader. The resulting values were then calculated relative to the absorbance found on the untreated control scaffold. The data is shown as a percentage in comparison to the control. Culture medium with the WST-1 reagent and no cells served as the blank.

### 2.8. dsDNA Assay

A total of 5 scaffolds (N = 5) from 5 treatment types (untreated, 15 min gold and genipin, 1 h gold and genipin, 24 h gold and genipin, and 24 h genipin only) were seeded with L929 mouse connective tissue fibroblasts and incubated for 1, 3, 7 and 10 days. Following cell culture, the discs were removed from their wells, gently rinsed, frozen at 70 °C. Samples were then lyophilized and submerged in papin digest and incubated at 60 °C for 24 h. A Quant-iT PicoGreen double stranded DNA quantification assay (Thermo Fisher Scientific, Waltham, MA, USA) was used to determine the cellularity of the scaffold. 25 μL of each papain digested sample were added to a 48-well plate. 225 μL of TE buffer (10 mM Tris-HCl, 1 mM EDTA, pH 7.5) and 250 μL of 2 μg/mL of PicoGreen reagent was added to each well and the plate was incubated for 5 min. Sample fluorescence was read at 480 nm excitation/520 nm using a Tecan Safire II plate reader. A Lambda DNA standard curve was used to determine DNA concentrations for the experimental samples.

*2.9. Scanning Electron Microscopy*

SEM was utilized to visualize the attachment of the fibroblasts on the scaffold and observe the overall microstructure. The scaffolds were either uncrosslinked or crosslinked with 3 mM genipin with or without the presence of 20 nm AuNP. Scaffolds were plated with fibroblasts for 1 or 3 days and were then prepared by fixation in 0.1 M cacodylate buffer containing 2% glutaraldehyde and 2% paraformaldehyde. Samples were critically point dried and examined using a FEI Quanta 600 F Environmental SEM.

*2.10. Statistical Analysis*

GraphPad Prism 8.0.1 (GraphPad Software, San Diego, CA, USA) was used to analyze experimental data. The Student's t test was used to analyze gold levels between samples with and without genipin. One-way analysis of variance was conducted followed by a Tukey-Kramer post-test to determine significant differences between means of the experimental groups. The results were considered statistically significant where *p* was less than 0.05.

## 3. Results

*3.1. Neutron Activation Analysis*

NAA was performed to measure the concentration of gold attached to the tissue scaffold. As shown in Table 1, the number of nanoparticles increased with the amount of crosslinking time with or without the use of genipin. For all time points, the scaffold with genipin had a higher concentration of gold, but the difference was only significant at the 15 min time point.

*3.2. Modulated Differential Scanning Calorimetry*

Figure 1 displays the denaturation temperatures of the tissue scaffold with different crosslinking times and different concentrations of genipin. The denaturation temperature significantly increases with only 15 min of genipin crosslinking. We see another significant increase at 24 h of crosslinking. Denaturation temperature also significantly increased with higher genipin concentrations.

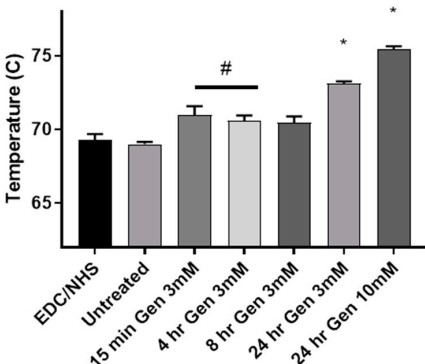

**Figure 1.** Results of modulated differential scanning calorimetry in which the mean denaturation temperature of each tissue is shown. * indicates significantly different from all other treatment types. # indicates significantly greater than the untreated scaffold (*p* < 0.05). Error bar displays standard error of the mean.

*3.3. Cell Viability*

Figure 2 shows the percent viability for each experimental group relative to decellularized untreated scaffolds (n = 5). The scaffolds crosslinked with genipin for 24 h, with or without gold nanoparticles, had significantly higher relative viability when compared to the uncrosslinked samples at day 1 and day 10. All other samples were not significantly higher than the uncrosslinked scaffolds.

**Table 1.** Neutron Activation Analysis Results. Concentration of gold (µg/g) on lyophilized tissue with and without the addition of genipin. * Indicates constructs that have significantly higher concentration of AuNPs than the sample incubated for the same amount of time without genipin. Values given as ± the standard deviation.

| Conjugation Time | Au | Au and Genipin |
|---|---|---|
| 15 min | 51 | 100 * |
| 1 h | 86 | 92 |
| 4 h | 153 | 190 |
| 8 h | 224 | 228 |
| 24 h | 373 | 389 |

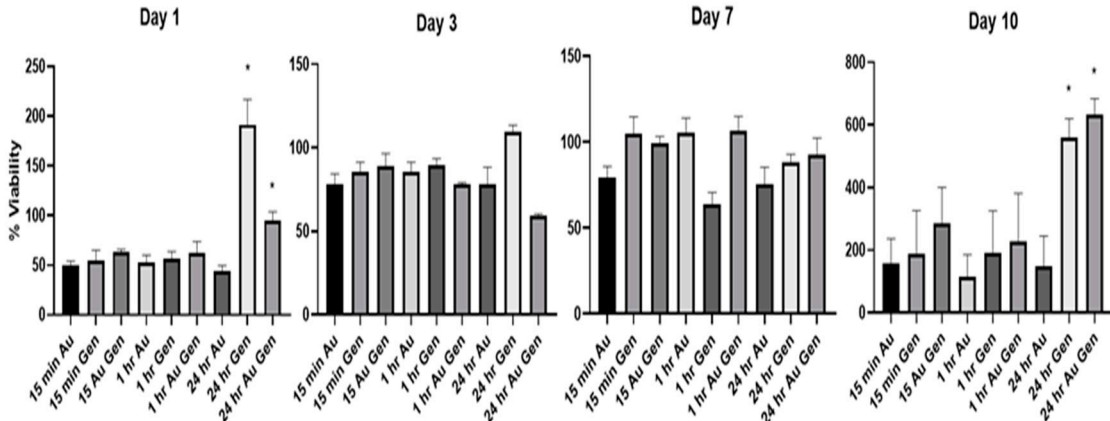

**Figure 2.** Cell proliferation reagent WST-1 assay results showing percent viability relative to the control, untreated tissue scaffolds. * indicates significantly different from untreated control ($p < 0.05$). Error bar displays standard error of the mean.

### 3.4. dsDNA Assay

The PicoGreen assay used to determine dsDNA content of the cells attached to the tissue scaffold and is shown in Figure 3. The sample with the 24 h genipin crosslinking with no gold nanoparticles had the highest levels of dsDNA for all time points. It was significantly higher than all other samples except the 24 h genipin with gold at day 3 and significantly higher than the samples crosslinked for 15 min or 1 h at day 1 and day 10.

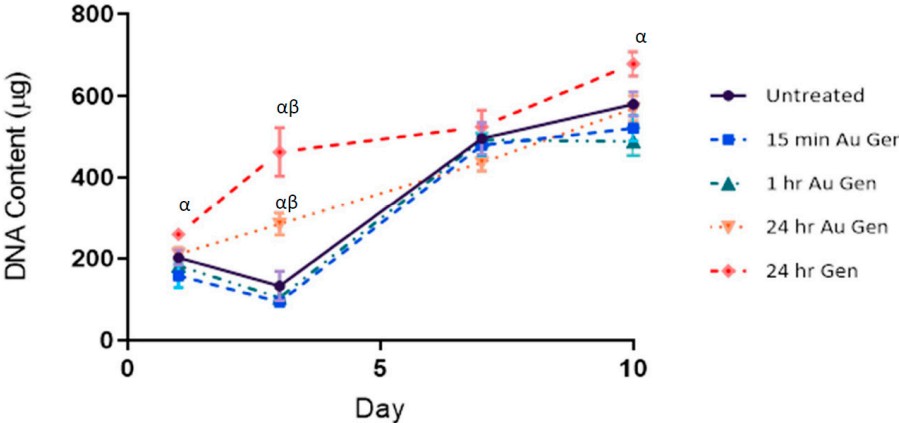

**Figure 3.** DNA content on scaffolds plated with L929 mouse fibroblasts for 1 to 10 days. ($\alpha$) indicates significantly higher DNA content than 15 min or 1 h crosslinking samples. ($\beta$) indicates significantly higher DNA content than the untreated samples. ($p < 0.05$). Error bar displays standard error of the mean.

### 3.5. Scanning Electron Microscopy

The SEM images clearly demonstrate the presence of fibroblast cells on the scaffold. The images also show the cells plated on the scaffolds with genipin alone are flusher, and more tightly adhered to the scaffold in comparison to the more spherical cells on the uncrosslinked scaffold at both day 1 and day 3. This is especially visible at the day 3 timepoint and can be seen on images D and G versus E and I in Figure 4. The fibroblasts on the genipin and AuNP scaffolds are moderately leveled but remained more spherical than the fibroblasts attached to scaffold with genipin alone as seen in images B, F and H.

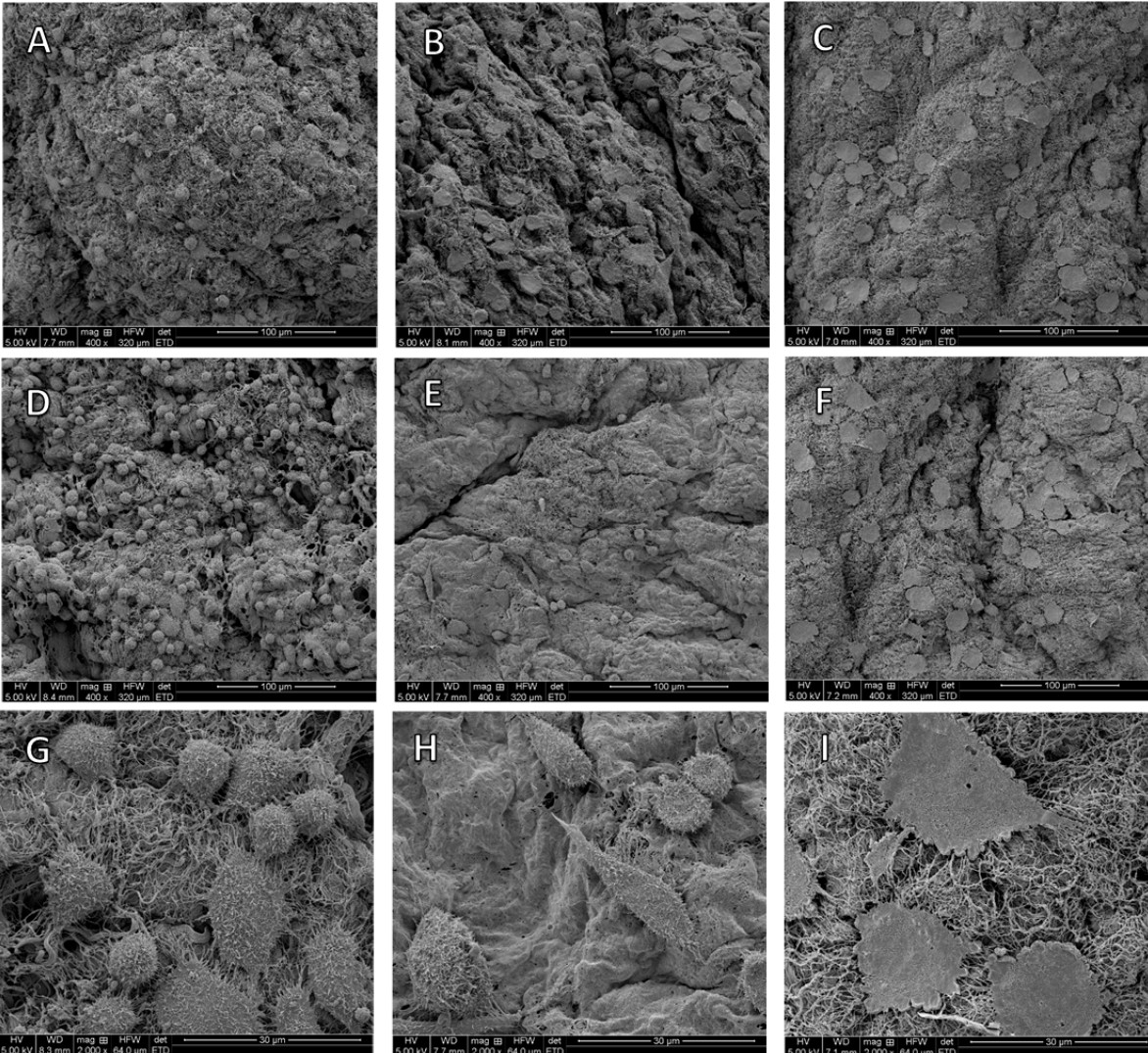

**Figure 4.** Scanning electron microscopy images taken 1 and 3 days after fibroblasts were plated on the of the scaffold (**A**). Uncrosslinked samples plated for 1 day (**B**). 24 h crosslinking with 3 mM genipin and 20 nm gold nanoparticles plated for 1 day (**C**). 24 h crosslinking with 3 mM genipin plated for 1 day without nanoparticles (**D,G**). Uncrosslinked samples plated for 3 days (**E,H**). 24 h crosslinking with 3 mM genipin and 20 nm gold nanoparticles plated for 3 days (**F,I**). 24 h crosslinking with 3 mM genipin without nanoparticles plated for 3 days. Images (**G–I**) feature the same treatment type as (**D–F**) respectively, but with increased magnification for clarity.

## 4. Discussion

Genipin was first utilized as a crosslinker in 1988 and since then its use has been studied for a wide array of applications [29]. In materials as diverse as the trachea, pericardium, cartilage, chitosan and gelatin gels, genipin has been successfully utilized to increase the mechanical strength and reduce

antigenicity [8,30–33]. Our study correlates with others on the ability of genipin to attach functionalized AuNPs to the tissue. The results of the study demonstrated the ability of genipin to conjugate gold nanoparticles to a decellularized porcine tissue scaffold while also maintaining good biocompatibility. The gold nanoparticles, functionalized with amino groups, allowed the genipin to covalently cross-link between the amino residues on the nanoparticles and the amino groups on the tissue. The modified cyclic form of genipin resides stably within the extracellular collagen matrix adding bridges from adjacent fibers to the functionalized AuNPs [34].

The NAA results demonstrated a correlation between the crosslinking time and the amount of AuNPs attached to the scaffold. An interesting result is noted in that there is an increase in the amount of AuNPs from 15 min to the 4 h immersion times. However, there is no significant increase from the 4 h to 8 h time point followed by a significant increase at 24 h. These biphasic results can be explained via the mechanism of genipin crosslinking. As explained by Butler et al., genipin crosslinking occurs via two separate reactions involving different sites on the genipin molecule. The first reaction is a nucleophilic attack of the genipin C3 carbon atom from a primary amine group which occurs almost immediately. The second slower reaction is the nucleophilic substitution of genipin's ester group to form a secondary amide [34]. The results clearly demonstrated genipin's biphasic reaction. In addition, the results also demonstrated that the functionalized AuNPs will bind to the tissue without the use of a crosslinker; however, the amount is significantly lower.

The DSC results provided additional confirmation on the binding ability of genipin. As shown in Figure 1, genipin demonstrated higher denaturation temperatures with the increased crosslinking times (15 min vs. 24 h) and with the higher genipin concentration (3 mM vs. 10 mM). The results also demonstrated the "two stage" binding ability of genipin in which the 4 h and 8 h crosslinking times displayed very similar denaturation temperatures. These results correlated with what other researchers reported. L. Bi et al. utilized genipin to crosslink a collagen chitosan scaffold. The results showed that the longer crosslinking times and higher genipin concentrations led to increased mechanical strength in their scaffold. However, it was also seen that when genipin concentrations reached above 1% (*w/v*) the mechanical strength decreased [35].

Our biocompatibility results demonstrated that genipin is not cytotoxic as confirmed by previous results [36]. On the contrary, the scaffolds with longer incubation times showed both an increase in cell numbers (Figure 3) and cell proliferation (Figure 2) when compared to untreated samples. We saw this result with or without the addition of the AuNPs meaning the genipin alone is associated with this increased cell growth. The enhanced cellular viability was demonstrated almost immediately with both biocompatibility assays showing a difference between the controls and the genipin treated scaffolds as early as 24 h after plating the cells. This led us to believe the enhanced cellular capabilities are the results of improved cell adhesion. To confirm this, SEM images were acquired at day 1 and day 3 after the scaffolds were plated with fibroblast cells. The cell plated on the scaffolds crosslinked with genipin alone are undoubtedly flusher to the scaffold than the other two treatment types. The treatment of genipin allows the cells to more quickly adhere and regain their spindle like appearance in comparison to the scaffolds left uncrosslinked or the scaffolds treated with genipin and gold.

Cell adhesion is a dynamic process involving interactions between cell cytoskeleton, extracellular matrix proteins, and peripheral membrane proteins. These adhesion protein complexes are crucial for the assembly of individual cells into the three-dimensional tissues and play an important role in further cell proliferation, viability, and differentiation [37]. It is well documented that physical surface properties, including stiffness, can significantly influence cell attachment. Forces generated by the cytoskeleton are applied to membrane attachment sites. This can deform materials that lack a degree of stiffness but cannot move an attachment site on a rigid surface. Consequently, cell morphology and functions hinge on substrate stiffness [38]. Gao et al. previously found that the use of genipin crosslinking increased surface roughness and stiffness on a hydrogel surface. This in turn resulted in better cell attachment, and better cell adhesion was associated with higher cell viability and proliferation [39]. As shown in Figure 2, there was an abrupt increase in 10 days 24 h Gen and 24 h Au

Gen samples. In correlation, the DSC results showed the samples immersed in the genipin solution for 24 h had a higher degree of crosslinking. This crosslinking most likely resulted in increased scaffold stiffness, and this may have contributed to the increased cell growth at 24 h as shown in Figure 2.

There is evidence that the improved cell attachment may be the result of direct genipin interactions with the cells. The switch from spherical to flattened shapes was not only demonstrated on the fibroblasts growing on the scaffolds, but this phenomenon was also witnessed in the cells attached to the well plate directly adjacent to the scaffolds crosslinked with genipin for 24 h. This is most likely the result of leaching of genipin, or other products of the crosslinking reaction, from the scaffold to the nearby cells. The exact mechanism for this is unclear, and there were no other cases of this phenomena cited in the literature. On the contrary, Wang et al. hypothesized that genipin may impair cell adhesion as it halved the mRNA expression of essential cell adhesion protein integrin β1 in chondrocytes [40]. However, our results clearly demonstrate genipin supporting fibroblasts as they attach, spread out, and flatten both on the scaffold and neighboring to it.

## 5. Conclusions

In this study, genipin was evaluated for its ability to attach gold nanoparticles to a decellularized porcine tendon scaffold. Genipin was able to attach gold levels equivalent to previously used methods in as little as 15 min. We also found cysteamine conjugated gold nanoparticles will attach to decellularized tissue without the use of any other compounds. The use of genipin resulted in no signs of cytotoxicity and instead accelerated cell attachment and growth when crosslinked for 24 h. Based on these results, genipin and gold nanoparticles can be used on tendon and ligament allografts to potentially decrease inflammation, improve cellular integration, and delay degradation. Our work focused on tendon tissues, but as genipin will bind to any amine group, there is no foreseeable reason why this would not also work on any type of decellularized tissue graft with similar results.

**Author Contributions:** Conceptualization, J.B. and S.A.G.; Data curation, M.B. and C.D.; Methodology, C.S.; Supervision, D.G. and S.A.G.; Writing—original draft, M.B.

**Funding:** This research was sponsored in part by the University of Missouri Enhanced Infrastructure for Biomedical Researchers Using Animal Models Grant and the University of Missouri Department of Biomedical, Biological, Chemical Engineering. The authors would like to thank Sherrie Neff and Justin Wilson for all their help.

**Conflicts of Interest:** The authors declare no conflict of interest.

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
