# Peer review of "Genipin Attachment of Conjugated Gold Nanoparticles to a Decellularized Tissue Scaffold"

_applsci, doi:10.3390/app9235231_

Round 1
Reviewer 1 Report
The present manuscript describes the method of conjugating gold nanoparticles to a decellularized porcine tissue scaffold using genipin crosslinker. The study along with the findings is interesting however there are certain issues that need to be addressed before possible publication.
What are the original, novelty or unique ideas behind this research as compared to previous research/other reported work? Why it is worth to know? Although the authors used a reported decellularization method but did they confirm that the matrix is completely free of cells? Also, how do they make sure that the matrix proteins do not lose their native structure after treatment with 70% ethanol? Fig.2. Please check the label of the y-axis label. Is it % viability or absorbance values? Fig.2. Why did the cell biocompatibility abruptly increased in 10 days 24 hr Gen and 24 hr Au Gen samples whereas it was significantly low until 7 days of studies? Need to justify. Fig. 3: Label for y-axis is missing. The image is very confusing and does not deliver the results clearly. What does an open square symbol represent? It is not mentioned in the image. I recommend re-drawing for the better representation of this fig. Please check and correct the order of images in Fig.4. Also describe what are F and I in the image.Author Response
Please see the attachment

Reviewer 2 Report
Mitch Bellrichard et al demonstrated that genipin can successfully and reproducibility attached gold nanoparticles to tissue in as little as 15 minutes without causing any cytotoxic effects. It can improved fibroblast attachment and proliferation. Author suggested that genipin can be used to attach gold nanoparticles to tissue in a fast, cell safe manner.
1. Please re-write the abstract. Author should write the rationale about why they chose genipin for gold nanoparticles.
2. Did author look into or tested other method of decellularization.
3. Sterilization process with ethanol does not sound sufficient for long term culture.
4. Cell proliferation reagent WST-1 assay should be normalized with total cell number. Please check the paper –
A. Satyam G. S. Subramanian M. Raghunath A. Pandit D. I. Zeugolis. In vitro evaluation of Ficoll‐enriched and genipin‐stabilised collagen scaffolds. Journal of tissue engineering and regenerative medicine Volume8, Issue3 March 2014 Pages 233-24.
5. Conclusion is very general.
6. Reference should be in uniform style. Please check all the references and follow the style guideline suggested by journal.
Reviewer 3 Report
The authors describe the use of gold nanoparticles crosslinked by genipin to decellularized porcine diaphragms.
Only in the title it is mentioned the aim for these biomaterials. In the manuscript it is never written, which sounds that it can be used for every tissue. It would be advisable to target this type of material to a specific tissue.
The materials and methods section are very confused. The treatment groups are not appropriate because there no treatment performed, but experimental groups instead, and the way that the groups are described is disorganized and also confused. T
The crosslinking procedure should be above the experimental groups and the Modulated DSC that appear from nowhere, probably should be in italic too, like all the other characterization techniques. In line 126, it is mentioned 10mM of genipin that it is not described in the experimental groups.
In the cell culture description, besides only describing the cells, it should be also mentioned the seeding technique as well as the culture time onto the materials. More details should be provided. Not after in the biocompatibility, that is in fact cell viability assessment. Overall the materials and methods section should be divided differently and re-written.
In the results section, table 1 is confuse and not in a presentable format. In Figure 2, the asterisks are not in the correct place. In Figure 3 the description of Y axis is missing.
Figure 4 should present all images at the same magnification. This is not correct.
The discussion is the best section in the paper.
The conclusions are ambiguous, because there is no application for these materials, despite what is mentioned in the title of the paper.
Reviewer 4 Report
The manuscript „Genipin Attachment of Conjugated Gold Nanoparticles Attachment to Ligament Scaffolds” presents new important findings in revealing the ability of genipin to attach cysteamine functionalized 20 nm gold nanoparticles to decellularized porcine tissue scaffold. The MS is suitable for publication after mandatory revisions:
In the ’Materials and Methods’ chapter: Is the subsection called ’Preparation of Scaffolds’ uncomplited or is this a larger group chapter title which includes the ’Tissue harvest and decellularization’, ’Treatment groups’, and ’Crosslinking’ sections? Please make it clear. The origin of the genipin should be stated. Do the authors produce themselves or do they buy it from somewhere? Please comment on this. Why the 20 nm diameter gold nanoparticles were chosen for the measurements? The particle size can be important in these kind of measurements. The authors should consult with and consider to cite the recent article of gold nanoparticles (and their size), titled „Interaction of Positively Charged Gold Nanoparticles with Cancer Cells Monitored by an in Situ Label-Free Optical Biosensor and Transmission Electron Microscopy”
(https://doi.org/10.1021/acsami.8b01546).
I suggest to replace the ’Crosslinking’ subsection with a new one in the ’Materials and Methods’ chapter, with the title of ’Genipin, gold nanoparticles and the crosslinking procedure’ (for insatnce). Here, the genipin (its origin), the nanoparticles (its origin; Sigma-Aldrich, Ted Pella, and the funcionalization of nanoparticles) and the crosslinking procedure should be summarized and written in details. The ’Sterilization’ procedure can be also mentioned here. This subsection should be after the ’Tissue harvest and decellularization’ subsection. The ’Modulated Differential Scanning Calorimetry’ should be written in italic, because this is a subsection, and it should be inserted after the ’Neutron Activation Analysis’ subsection. The Table 1 should be in an other form. It would be clearer with 3 coloumns: the first is the Incubation time, the second is the Au and the third is the Au+genipin. Page 5, line 201: Instead of „The samples denatuation time” it should be „The denaturation time of the samples”. Caption of Figure 1: the α,β,γ definitions are not very understandable. When the word ’significant’ is used its exact meaning shouls be clear. Similar as in Figure 3. Please make it clear. Figure 3. The curves should be coloured for better visibility. In the ’Scanning electron microscopy’ section (page 7), the Figure 4 or its parts (A, B, C…) are not referred in the paragraph.In the Caption of Figure 4: The last sentence is unfinished. What is F and I? They are missing. Page 7, line 251. „collagen matric” ? Page 8, line 261. „…genipin’s the ester group…”, correctly: „…genipin’s ester group…” Page 8, line 270. „…others researhers…”, correctly: „…other researhers…” Page 8, line 273. „We did not test this high of genipin concentration in our study.” This sentence is unnecessary. The reference style is not adequate for the Applied Sciences journal, correct it to the right style. A new figure of the genipin cross-link between the amino residues should be inserted to the ’Genipin, gold nanoparticles and the crosslinking procedure’ subsection or into the ’Introduction’.Author Response
Please see the attachment

Round 2
Reviewer 1 Report
The authors have now incorporated the required changes as recommended and justified all the issues raised. It can be accepted for publication in Applied Sciences.
Author Response
Thank you for taking the time to review our submission. Your feedback is much appreciated.